# Bragg Grating Assisted Sagnac Interferometer in SiO_2_-Al_2_O_3_-La_2_O_3_ Polarization-Maintaining Fiber for Strain–Temperature Discrimination

**DOI:** 10.3390/s20174772

**Published:** 2020-08-24

**Authors:** Zhifang Wu, Peili Wu, Maryna Kudinova, Hailiang Zhang, Perry Ping Shum, Xuguang Shao, Georges Humbert, Jean-Louis Auguste, Xuan Quyen Dinh, Jixiong Pu

**Affiliations:** 1Fujian Key Laboratory of Light Propagation and Transformation, College of Information Science and Engineering, Huaqiao University, Xiamen 361021, China; jixiong@hqu.edu.cn; 2School of Electrical and Electronic Engineering, Nanyang Technological University, 50 Nanyang Avenue, Singapore 639798, Singapore; WUPE0004@e.ntu.edu.sg (P.W.); hzhang023@e.ntu.edu.sg (H.Z.); shum@ieee.org (P.P.S.); XGshao@ntu.edu.sg (X.S.); XDDinh@ntu.edu.sg (X.Q.D.); 3CINTRA CNRS/NTU/THALES, UMI 3288, 50 Nanyang Drive, Singapore 637553, Singapore; 4XLIM Research Institute, UMR 7252 CNRS/University of Limoges, 123 Avenue Albert Thomas, 87060 Limoges, France; marynakudinova@gmail.com (M.K.); georges.humbert@xlim.fr (G.H.); jean-louis.auguste@xlim.fr (J.-L.A.); 5Thales Solutions Asia Pte Ltd., R&T Department, 21 Changi North Rise, Singapore 498788, Singapore

**Keywords:** lanthanum-aluminum silicate glass, polarization-maintaining fiber, fiber Bragg grating, Sagnac interferometer

## Abstract

Polarization-maintaining fibers (PMFs) have always received great attention in fiber optic communication systems and components which are sensitive to polarization. Moreover, they are widely applied for high-accuracy detection and sensing devices, such as fiber gyroscope, electric/magnetic sensors, multi-parameter sensors, and so on. Here, we demonstrated the combination of a fiber Bragg grating (FBG) and Sagnac interference in the same section of a new type of PANDA-structure PMF for the simultaneous measurement of axial strain and temperature. This specialty PMF features two stress-applied parts made of lanthanum-aluminum co-doped silicate (SiO_2_-Al_2_O_3_-La_2_O_3_, SAL) glass, which has a higher thermal expansion coefficient than borosilicate glass used commonly in commercial PMFs. Furthermore, the FBG inscribed in this SAL PMF not only aids the device in discriminating strain and temperature, but also calibrates the phase birefringence of the SAL PMF more precisely thanks to the much narrower bandwidth of grating peaks. By analyzing the variation of wavelength interval between two FBG peaks, the underlying mechanism of the phase birefringence responding to temperature and strain is revealed. It explains exactly the sensing behavior of the SAL PMF based Sagnac interference dip. A numerical simulation on the SAL PMF’s internal stress and consequent modal effective refractive indices was performed to double confirm the calibration of fiber’s phase birefringence.

## 1. Introduction

High-birefringence (Hi-Bi) fibers, a category of polarization-maintaining fibers (PMFs), refer to those having significant effective refractive index difference between their two orthogonally polarized fundamental modes [1,2]. Thanks to their special capability of maintaining the polarization state of transmitted light, PMFs play very important roles in optical communication and sensing systems. Especially, PMFs provide unique convenience for constructing fiber-based Sagnac interferometers, which have been demonstrated with great success in optical fiber gyroscopes, comb filters and other high-accuracy measurements of current, vibration, strain, temperature and so on [3,4,5,6]. Besides, by combining with fiber gratings [7], Fabry–Perot interference [8], Mach–Zehnder interference [5], or microresonators [9], PMF-based Sagnac interferometers have been exploited for multi-parameter sensing applications as well.

Generally, there are two approaches to break the degeneracy of two fundamental modes and enlarge their effective refractive index difference considerably. The first one basically consists of designing and fabricating noncircular-shape-core fiber [10,11]. For example, elliptical-core fibers were fabricated and showed appreciable birefringence as early as 1970s [12,13]. Benefiting from the development of microstructured optical fibers (MOFs) in 1990s [14,15], lots of very high-birefringence MOFs were demonstrated with various sizes of air holes [16,17]. The second approach to generating stable birefringence in optical fibers is essentially related to elasto-optic effect. By replacing some parts of cladding with a high-expansion glass, asymmetric stress will be frozen in the fiber due to the different thermal expansion coefficients between the replacement part and the rest of cladding as the fiber is drawn and cooled down. Such stress in turn creates a significant effective refractive index difference between the core modes, which are parallel and perpendicular to the stress direction through elasto-optic effect, resulting in a linear birefringence in the drawn fiber. The areas filled with high-expansion glass are called stress-applying parts (SAPs). The classic geometry of SAPs includes bow-tie, PANDA, and elliptical inner cladding [18].

Considering the thermo-mechanical compatibility between SAPs and pure silica cladding, most SAPs are made of borosilicate glass whose thermal expansion coefficient (∼$1.0\times 10^{-6}$ K^−1^) is about two times higher than that of pure silica ($5.4\times 10^{-7}$ K^−1^) [19,20]. Recently, a new kind of lanthanum-aluminum co-doped silicate (SiO_2_-Al_2_O_3_-La_2_O_3_, SAL) glass was used as SAPs in a PANDA-type PMF [20,21]. The thermal expansion coefficient of this SAL glass was calibrated to be around $5.32\times 10^{-6}$ K^−1^, which is almost ten times that of silica and five times higher than that of borosilicate glass. The fabricated SAL PMF showed comparable phase birefringence with commercial PANDA PMFs with borosilicate SAPs.

In this paper, we demonstrated for the first time the integration of Bragg grating into an SAL PMF and its sensing capabilities when it was inserted in a Sagnac interferometer. At the same time, the phase birefringence of the SAL PMF was further calibrated more precisely by calculating the difference between the Bragg peaks of two orthogonal-polarization modes. The distinct responses of the Bragg grating peaks and interference fringe were applied in the discrimination of axial strain and temperature. The underlying mechanism of the behavior of the Sagnac interference fringe was analyzed as well.

## 2. Materials and Methods

The specialty PMF used in this work was conceived and fabricated at XLIM research institute France by using powder-in-tube technology [20]. The scanned electronic microscopy (SEM) picture in Figure 1a shows that the fiber has PANDA geometry, consisting of a germania-doped silica core and two flanked stress-applied rods made of SAL glass. Quantitatively, the latter one contains 70 mol% SiO_2_, 20 mol% Al_2_O_3_ and 10 mol% La_2_O_3_. The diameters of the core and cladding are of 8.4 μm and 140 μm, respectively. The centers of two stress rods are about 36.5 μm away from the core center, and their diameters are about 25.0 μm and 25.2 μm, respectively.

The corresponding refractive index profile of this fiber was measured by an IFA-100 Fibre Index Profiler and then plotted in Figure 1b. The core has a quasi-step-index profile and the maximum refractive index difference from that of pure silica is around 0.0062. The refractive indices of two SAL rods show parabolic-like distribution and the maximum values are about 0.0428 and 0.0504, respectively. Contrary to conventional PANDA PMFs whose stress rods are made of borosilicate, in our fiber, the refractive indices of SAPs are higher than that of the fiber core since the dopant (La_2_O_3_) has much higher refractive index than pure silica [22]. Referring to conventional PANDA PMFs, the axis parallel to the applied-stress direction is defined as “slow axis”, whereas the perpendicular axis is defined as “fast axis” [1].

As a high-birefringence fiber, the SAL PMF can be employed to construct a Sagnac interferometer [3,23]. Meanwhile, since the core of the SAL PMF is doped with germanium, it is also suitable for functionalization by inscribing Bragg gratings. When the SAL PMF with the inscribed Bragg grating is inserted into a Sagnac loop mirror, the Sagnac interference and Bragg grating resonance are combined together, as shown in Figure 2. It is noteworthy that the configuration is different from those in previous works [7,24], in which the grating is inscribed in standard single-mode fiber (SMF) and then connected with PMFs. The spatial distance between the grating and PMF may cause the device to respond to external variation from different places, especially in sensing mechanical quantities. In our scheme, however, the FBGs are integrated inside the SAL PMF directly. Thus, there is almost no spatial distance between the FBGs and PMF, indicating that they respond to the perturbation from same place.

The Bragg grating in this SAL PMF was fabricated by using conventional ultraviolet (UV) laser lateral exposure method, in which the fiber was exposed to the interference fringe by scanning a phase mask with a UV laser [25]. Before being exposed to UV light, the photosensitivity of SAL PMF was enhanced via hydrogen loading process at 12 Mpa and 80 °C for one week. Next, a 56-mm-long SAL PMF was spliced with two single-mode fibers and then connected to a broadband source (BBS, Infinon Research Broadband Source) and an optical spectrum analyzer (OSA, Yokogawa AQ6370C) through a 50:50 coupler, as shown as the schematic diagram in Figure 2. A polarization controller was inserted into the fiber loop to optimize the output interference spectrum. The UV light was generated from a frequency-doubled continuous-wave (CW) Argon laser (Coherent, INNOVA 90C) and the power was about 70 mW. The period of phase mask was 1070.20 nm, corresponding to the grating pitch of 535.10 nm. The resolution of OSA was set to 0.02 nm. Before laser scanning, an additional alignment of the refractive index axis was conducted to avoid that the core was shadowed by the stress rods. The UV laser was moved out of the mask region and focused on the SAL PMF. Since the diffraction pattern illuminated through the fast axis was significantly different from that through the slow axis, the illumination direction was aligned along the fast axis by rotating the fiber and monitoring the diffraction pattern. Finally, the Bragg grating was inscribed by moving the UV laser to scan the phase mask with the speed of 0.02 mm/s. The inscription length was 10.00 mm.

## 3. Results and Discussions

### 3.1. Transmission Spectra of Fabricated Device

After the grating inscription process finished, the transmission spectrum was measured and plotted in Figure 3. As shown, two Bragg grating peaks, corresponding to the Bragg reflections of slow-axis and fast-axis polarized modes, are embedded in the interference fringe and their central wavelengths are 1549.62 nm and 1549.27 nm, respectively. According to the phase-matching condition of fiber Bragg grating, $\lambda_{\mathrm{res}}=2n_{\mathrm{eff}}\cdot \Lambda$, the effective refractive indices of slow-axis and fast-axis polarized modes ($n_{\mathrm{eff},s}$ and $n_{\mathrm{eff},f}$) can be calculated, respectively.

The phase birefringence *B*, defined as the effective refractive index difference between slow-axis and fast-axis polarized modes, can be derived as follows:
(1)$$\begin{array}{c} B=n_{\mathrm{eff},s}-n_{\mathrm{eff},f}=\frac{\left(\lambda_{\mathrm{res},s}-\lambda_{\mathrm{res},f}\right)}{2\Lambda }, \end{array}$$
where $\lambda_{\mathrm{res},s}$ and $\lambda_{\mathrm{res},f}$ are the resonant wavelengths of slow-axis and fast-axis polarized modes, respectively. For the fabricated Bragg grating in the SAL PMF, Λ=535.10 nm, $\lambda_{\mathrm{res},s}=1549.62$ nm and $\lambda_{\mathrm{res},f}=1549.27$ nm, the phase birefringence *B* of this SAL PMF is characterized as $3.27\times 10^{-4}$. It matches well with the previously reported results [20]. Moreover, this characterization result should be more precise since the bandwidth of Bragg grating peaks are much narrower than that of interference fringe. The corresponding beat length ($L_{b}=\lambda /B$) is around 4.74 mm, which is comparable with that of commercial PANDA PMF [1].

### 3.2. Stress and Birefringence Simulation

Since the birefringence in the SAL PMF is induced by the asymmetric thermal stress through elasto-optical effect, the internal thermal stress of the fiber and the consequent phase birefringence were numerically analyzed by using the solid mechanics and wave optics modules in COMSOL Multiphysics [26,27]. The detailed parameters of the fiber’s structure and materials are listed in Table 1. In order to simplify the simulation model, the fiber’s structure was defined to be axial symmetric along the fast and slow axes. Moreover, the core, SAL rods and outer cladding were all drawn by perfectly circular shape, and they were of 8.4 μm, 24.2 μm and 140 μm in diameter, respectively. A step-index profile of the refractive index was adopted. The material refractive indices of the pure silica and germania-doped silica (core region) were described by a three-term Sellmeier dispersion formula with corresponding coefficients [28]. The mole fraction of germania in fiber’s core can be deduced by the measured refractive index difference of the core $\Delta n_{\mathrm{core}}$ and the relationship between the refractive index variation and the doping concentration of germania ($C_{{\mathrm{GeO}}_{2}}\left(\mathrm{mol.}\%\right)=\Delta n_{\mathrm{core}}/0.00146$) [29]. Here, $\Delta n_{\mathrm{core}}$ equals 0.0062 according to measured index contrast shown in Figure 1b. The index of SAL glass was represented by adding a fixed value to the dispersion formula of pure silica. The fixed value was the average index difference of the SAL rods compared to the pure silica cladding. It was estimated around 0.045 based on the measured index distribution. Since the doping concentration of the fiber’s core was only around 4.0 mol.%, the core was assumed to have the same Young’s modulus, Poisson’s ratio and relative density as those of pure silica, while those of SAL rods were much higher and cited from previous work [20]. All the materials were set with the same stress-optical coefficients.

When the fiber was cooled down from 1100 °C (reference temperature) to 20 °C (operating temperature), the difference between the stress distribution along slow-axis and fast-axis directions was simulated and illustrated in Figure 4. As shown, the mean stress distributes anisotropically on the SAL PMF’s cross section, and the mean stress near the core region is around $10^{8}$ Pa. Consequently, the material refractive indices become anisotropic accordingly, resulted in the breakage of the degenerateness of fiber’s fundamental modes. Figure 5a,b show the simulated electric field distribution of x-polarized and y-polarized fundamental modes, denoted respectively with LP${}_{01,X}$ and LP${}_{01,Y}$. Their polarization states (marked with white arrows) point to the slow axis and fast axis, respectively. Their effective refractive indices decrease monotonically from wavelengths of 1100 nm to 1900 nm, while the corresponding phase birefringence increases very slightly. The simulated phase birefringence is estimated at 3.6 × $10^{-4}$ for a wide wavelength range. It agrees basically with the experimental results (3.27 × $10^{-4}$). The slight gap may be attributed to the structural mismatch between the experimental fiber and simulation model.

### 3.3. Sensing Performance

The sensing performance of the proposed device was investigated as the following. Firstly, the sensor head was clamped on two translation stages whose fine resolution was limited to 0.5 μm. One stage was fixed, whereas the other one was driven by a micrometer to stretch fiber [25]. When the axial strain was increased from 0 to 1200 με, as shown in Figure 6a,b, both interference pattern and Bragg peaks shift to longer wavelengths. Moreover, it is worth noting that the red shift of two FBG peaks are very similar and the wavelength gap between two peaks becomes slightly larger with the increment of applied axial strain. It indicates that the phase birefringence of the SAL PMF is enlarged as the fiber is stretched increasingly. For PMF-based Sagnac interferometer, the relationship between the shift of interference wavelength ( $\Delta \lambda_{\mathrm{SI},\varepsilon }$) and applied axial strain (ε) can be described by $\Delta \lambda_{\mathrm{SI},\varepsilon }=\lambda_{\mathrm{SI}}\left(1+P_{e}\right)\varepsilon$, where $P_{e}=\Delta B\left(\varepsilon \right)/B$ is the strain-optic coefficient of the fiber’s birefringence. Since both physical length and birefringence depend on axial strain positively, the Sagnac interference spectrum will consequently shift to longer wavelengths.

By following the wavelength variations of the fast-axis Bragg peak and one interference dip, the strain sensitivities of the Bragg grating and the interference dip of the proposed device can be characterized to be 7.60 $\times 10^{-4}$ nm/μϵ and 1.00 $\times 10^{-2}$ nm/μϵ, respectively, as shown as the linear fitting curves in Figure 7. Their corresponding R-squared factors of linear fitting are 0.9996 and 0.9921, respectively.

Then, the sensor head was placed into a high-accuracy column oven to characterize its thermal performance. The temperature precision of this oven is 0.1 °C. A tiny tension was applied at both ends of the sensor head to keep it straight and then the temperature was increased from 25 °C to 85 °C with the step of 5 °C. The corresponding transmission spectra of the Sagnac interference and Bragg grating were recorded and illustrated in Figure 8a,b, respectively. As shown, the interference spectrum shifts to shorter wavelengths with the increment of temperature, while the Bragg peaks shift to longer wavelengths. Moreover, the gap between two Bragg peaks decreased apparently in the process of rising temperature. It means that the phase birefringence of the SAL PMF tends downwards when fiber temperature goes upwards. Similarly, the variation of interference wavelength ($\Delta \lambda_{\mathrm{SI},T}$) is related to fiber’s thermal expansion and the birefringence change caused by thermo-optic effect, $\Delta \lambda_{\mathrm{SI},T}=\lambda_{\mathrm{SI}}\left(\alpha +\Gamma \right)T$, where α is the thermal expansion coefficient and Γ is the equivalent thermo-optic coefficient of birefringence. As illustrated in Figure 8b, two FBG peaks goes closer with increasing temperature. It indicates that the difference between the effective refractive indices of two polarized modes (i.e., phase birefringence) becomes smaller and the equivalent thermo-optic coefficient of birefringence Γ should be negative. Although α is positive, the thermo-optic effect induced birefringence variation contributes dominantly to the shift of interference fringe [6]. Therefore, the negative response of the interference dip to temperature, shown in Figure 8a, can be well explained.

In parallel with the strain measurement, the temperature sensitivities of the fast-axis polarization Bragg peak and the interference dip are taken at 1.18 × $10^{-2}$ nm/°C and −8.57 × $10^{-1}$ nm/°C, respectively, as shown by the linear fitting results in Figure 9. Their corresponding R-squared factors of linear fitting are of 0.9996 and 0.9921, respectively.

On the basis of the different responses of the fast-axis polarization Bragg peak and the Sagnac interference dip to axial strain and temperature, this proposed sensor can be applied to measure strain and temperature simultaneously by using the following matrix [30,31,32]:(2)$$\begin{array}{ccc} \left[\begin{array}{c} \Delta \varepsilon \\ \Delta T \end{array}\right] & = & {\left[\begin{array}{cc} S_{\varepsilon ,\mathrm{SI}} & S_{T,\mathrm{SI}}\\ S_{\varepsilon ,\mathrm{FBG}} & S_{T,\mathrm{FBG}} \end{array}\right]}^{-1}\left[\begin{array}{c} \Delta \lambda_{\mathrm{SI}}\\ \Delta \lambda_{\mathrm{FBG}} \end{array}\right]\\ & = & {\left[\begin{array}{cc} 1.00\times 10^{-2} & -8.57\times 10^{-1}\\ 7.60\times 10^{-4} & 1.18\times 10^{-2} \end{array}\right]}^{-1}\left[\begin{array}{c} \Delta \lambda_{\mathrm{SI}}\\ \Delta \lambda_{\mathrm{FBG}} \end{array}\right] \end{array}$$
where Δε and ΔT refer to strain and temperature variations; $\Delta \lambda_{\mathrm{SI}}$ and $\Delta \lambda_{\mathrm{FBG}}$ correspond to wavelength shifts of the Sagnac interference dip and the fast-axis polarization Bragg peak, respectively; $S_{\varepsilon ,\mathrm{SI}}$, $S_{T,\mathrm{SI}}$, $S_{\varepsilon ,\mathrm{FBG}}$, and $S_{T,\mathrm{FBG}}$ are the strain and temperature sensitivities of Sagnac interference dip and fast-axis polarization Bragg peak, respectively.

## 4. Conclusions

In conclusion, the Bragg grating fabrication in a new type of polarization-maintaining fiber (SAL PMF) whose stress-applied rods were made of SAL glass was reported for the first time, to the best of our knowledge. Based on this SAL PMF with the integrated FBG, a Sagnac interferometer was constructed and demonstrated for the simultaneous measurement of axial strain and temperature. The strain and temperature sensitivities of the proposed device reach 1.00 × $10^{-2}$ nm/με and −8.57 × $10^{-1}$ nm/°C, respectively. They are about 100 times and 85 times higher than those of normal FBG-based sensors, respectively [33]. Moreover, the Bragg peaks functioned as an important indicator for investigating the phase birefringence of the SAL PMF. Firstly, by calculating the effective refractive indices of two Bragg peaks, the SAL PMF’s phase birefringence was calibrated more precisely compared with the calculation from the interference spectrum, since the bandwidths of Bragg peaks were much narrower than those of the interference fringe [20]. Secondly, the variation of Bragg peaks’ separation reflected the tendency of SAL PMF’s birefringence with changing axial strain or temperature as well. Thus, it explained explicitly the response behavior of the Sagnac interference dip in the axial strain and temperature sensing measurements.

## Figures and Tables

**Figure 1 sensors-20-04772-f001:**
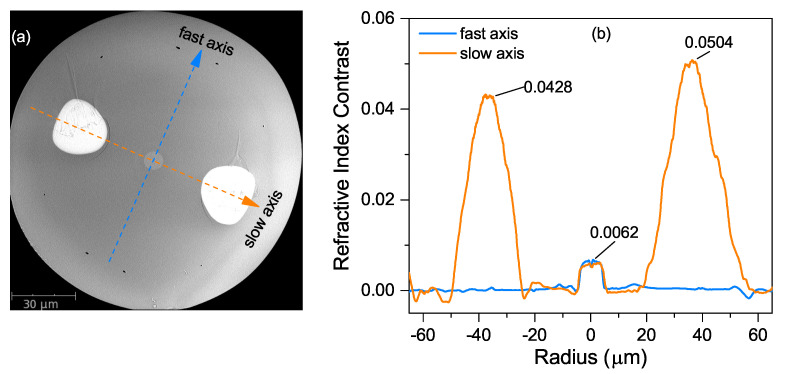
Scanning electronic microscopy picture (**a**) and corresponding refractive index profile (**b**) of the SiO_2_-Al_2_O_3_-La_2_O_3_ (SAL) polarization-maintaining fiber (PMF).

**Figure 2 sensors-20-04772-f002:**
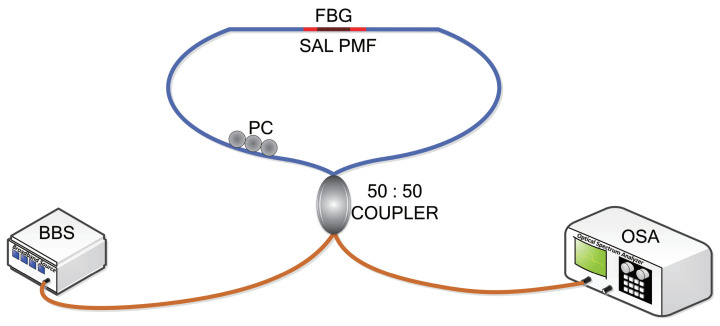
Schematic diagram of the proposed configuration and experimental setup. BBS: broadband source; OSA: optical spectrum analyzer; PC: polarization controller; FBG: fiber Bragg grating; SAL PMF: SiO_2_-Al_2_O_3_-La_2_O_3_ polarization-maintaining fiber.

**Figure 3 sensors-20-04772-f003:**
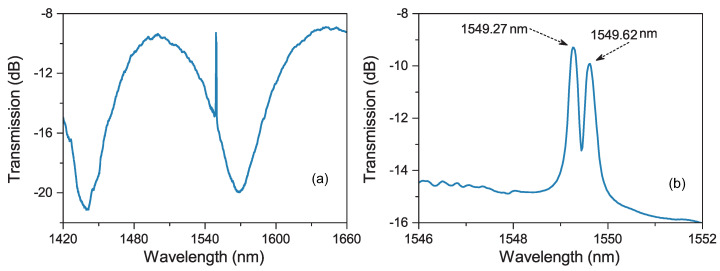
Spectra of the SAL PMF based Sagnac interference (**a**) and Bragg resonant peaks (**b**) after grating inscription.

**Figure 4 sensors-20-04772-f004:**
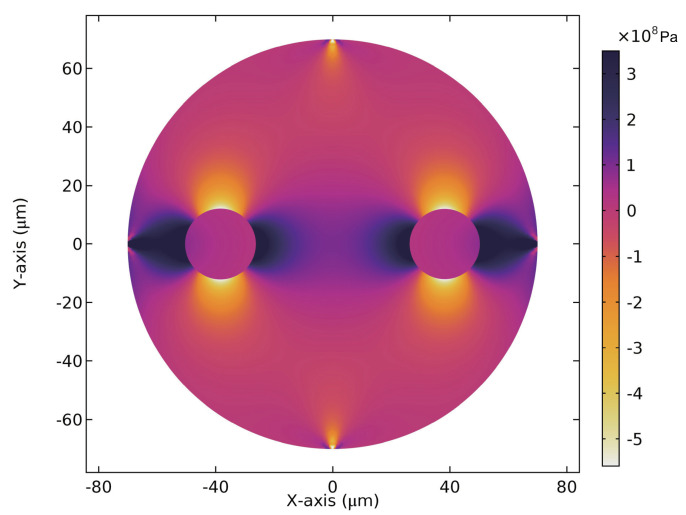
Thermal stress difference $\sigma_{x}-\sigma_{y}$ distribution of the SAL PMF cross section, where $\sigma_{x}$ and $\sigma_{y}$ are the stress distribution along the slow and fast axes, respectively.

**Figure 5 sensors-20-04772-f005:**
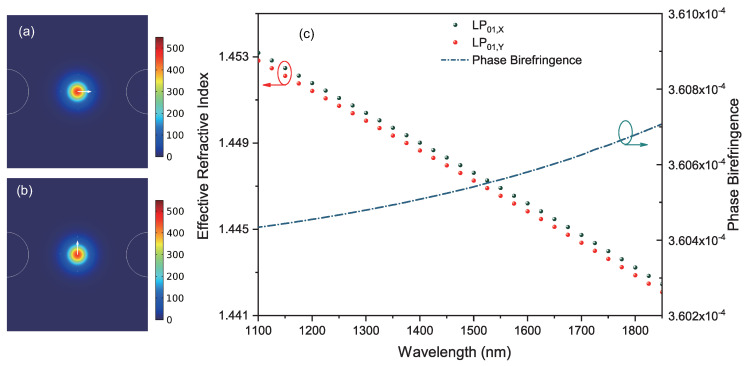
The electric field distribution of x-polarized fundamental mode (**a**) and y-polarized fundamental mode (**b**). (**c**) The effective refractive index variations of x-polarized (black sphere) and y-polarized (red sphere) modes in the wavelength range from 1100 nm to 1900 nm, and the dispersion curve of the corresponding phase birefringence (dark blue dot-dash line).

**Figure 6 sensors-20-04772-f006:**
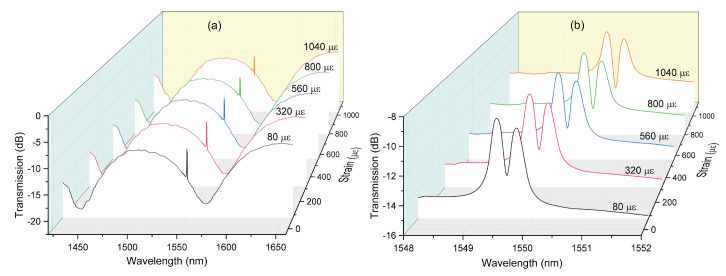
Spectral shifts of Sagnac interference dip (**a**) and Bragg resonant peaks (**b**) with varying axial strain.

**Figure 7 sensors-20-04772-f007:**
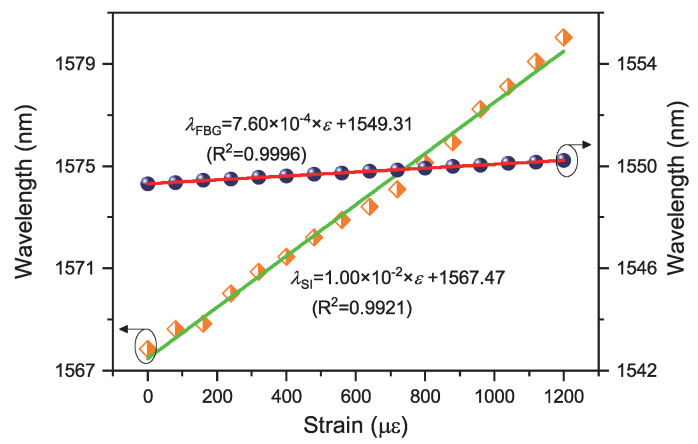
Axial-strain responses of Sagnac interference dip (orange square with green line) and Bragg peak of fast-axis polarized mode (blue sphere with red line).

**Figure 8 sensors-20-04772-f008:**
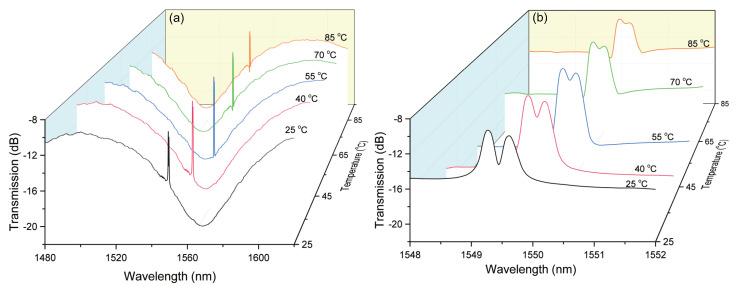
Spectral shifts of Sagnac interference dip (**a**) and Bragg resonant peaks (**b**) with varying temperature.

**Figure 9 sensors-20-04772-f009:**
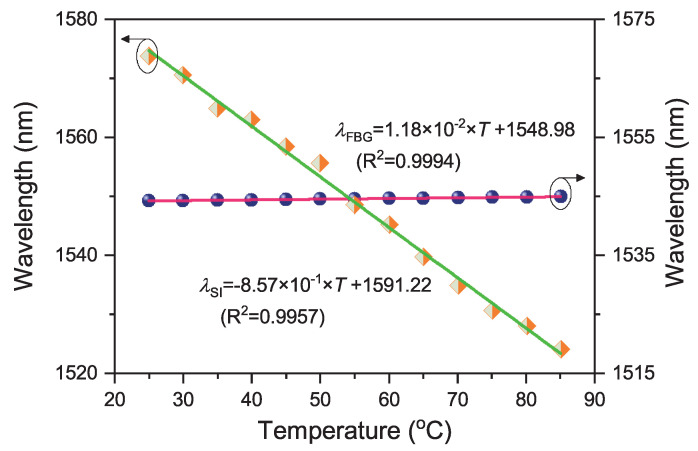
Thermal responses of the Sagnac interference dip (orange square with green line) and the Bragg peak of fast-axis polarized mode (blue sphere with red line).

**Table 1 sensors-20-04772-t001:** Structure and material parameters used in the simulation [20,26].

	Cladding	Core	SAL Rods
Material	pure silica	GeO_2_ doped silica	La_2_O_3_-Al_2_O_3_ doped silica
Diameter (μm)	140	8.4	24.2
Young’s modulus (GPa)	78.3	78.3	110.7
Poisson’s ratio	0.186	0.186	0.282
Relative density (kg/m^3^)	2203	2203	3346.7
Thermal expansion coefficient (K^−1^)	5.4 × $10^{-7}$	1.0 × $10^{-6}$	5.32 × $10^{-6}$
First stress optical coefficient ($m^{2}$/N)	0.65 × $10^{-12}$		
Second stress optical coefficient (m^2^/N)	4.2 × $10^{-12}$		
Operating temperature (°C)	20		
Reference temperature (°C)	1100		

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
