# Peer review of "Bragg Grating Assisted Sagnac Interferometer in SiO2-Al2O3-La2O3 Polarization-Maintaining Fiber for Strain–Temperature Discrimination"

_sensors, 2020, doi:10.3390/s20174772_

Round 1

Reviewer 1 Report

Overall Comments:

The reviewer commends the authors on their impressive combination of a SAL PMF with FBG. The manufacture of the fibre optic sensor is novel and interesting and the characterisation methods and results of the sensors interferometry and FBG components are logical and interesting. However, the application of the sensor is unclear, and the manuscript requires professional English and grammatical editing before it is ready for publication.

Major Comments:

The reviewer is confused about the application of this novel sensor. As the reviewer understands the default application of a Sagnac interferometer is to measure axial rotation of a device through changes in the relative path length between the split traveling beams – e.g strain does not play a role in it. While the reviewer can see the value of adding an FBG to the system to compensate for temperature changes, the impact of strain, which is explored extensively throughout the manuscript, is not clear, as presumably strain in a Sagnac interferometer would be 0.

Based on other literature, it appears that the Sagnac effect can be used to measure strain by having a fixed differential path length on a non-rotating frame. If the primary application of the sensor is to detect strain in such a way, the authors should justify (e.g higher accuracy, stability, sensitivity) the use of such a complex setup compared to a simple FBG strain sensor, which can already measure strain and temperature independently given an appropriate configuration.

Some discussion should be added on how this device compares (accuracy, sensitivity, response) against other strain/angular velocity/temperature sensors, whether they be MEMS, PMF, FBG or otherwise.

Minor Comments:

Abstract typo: magenatic should be magnetic.

Abstract define PANDA

Page 6 line 147/148 Results and discussion: for ease of comparison between experiments and simulation please restate the experimental birefrigerance value (3.27 x 10-4)

Reviewer 2 Report

This work proposes a sensor to measure simultaneously strain and temperature using a Sagnac interferometer with specialty Hi-Bi polarization-maintaining fiber and a Bragg grating into the cavity.

The manuscript is presented correctly and sounds technically correct. Although a new sensor scheme is proposed for simultaneous measurement of strain and temperature, its novelty is unclear. To complement this work, it is recommended to address the following issues.

  1. Include the advantages offered by the proposed sensor over similar schemes to measure simultaneously strain and temperature using Sagnac interferometers and Bragg gratings such as:

Frazão et al. Simultaneous measurement of multiparameters using a Sagnac interferometer with polarization maintaining side-hole fiber. Applied Optics, 2008.

[7] Zhou, D.P et al. Simultaneous measurement of strain and temperature based on a fiber Bragg grating combined with a high-birefringence fiber loop mirror. Opt. Commun. 2008.

Hyun-Min Kim et al. Simultaneous measurement of strain and temperature with high sensing accuracy. 14th OptoElectronics and Communications Conference. Hong Kong, 2009.

Huanhuan Yan et al. Dual parameter measurement system for temperature and stress based on Sagnac interferomter. Journal of the European Optical Society-Rapid Publications, 2020.

and so on…

  1. What advantages are there in the proposed sensor when using SiO2-Al2O3-La2O3 fiber in the cavity over commercial polarization-maintaining fibers (Panda or Bow-tie)?
  2. Explain more in detail the temperature negative sensitivity of the Sagnac interferometer with the specialty SiO2-Al2O3-La2O3 fiber.
  3. Add the PC term (polarization controller) in Figure 2.
  4. Add the Pa units in Figure 4.

According to the above, I believe that this work fulfills the standard requirements to be accepted in the Sensors Journal once the authors attend to the aforementioned recommendations.
